# Prevalence of Childhood Asthma and Allergies and Their Associations with Perinatal Exposure to Home Environmental Factors: A Cross-Sectional Study in Tianjin, China

**DOI:** 10.3390/ijerph18084131

**Published:** 2021-04-14

**Authors:** Agnes S. Ellie, Yuexia Sun, Jing Hou, Pan Wang, Qingnan Zhang, Jan Sundell

**Affiliations:** Tianjin Key Laboratory of Indoor Air Environmental Quality Control, School of Environmental Science and Engineering, Tianjin University, No.135 Yaguan Road, Haihe Education Park, Tianjin 300350, China; agnesellie61@gmail.com (A.S.E.); jing_houj@163.com (J.H.); panwang@tju.edu.cn (P.W.); qingnanstar@126.com (Q.Z.); sundellcc@gmail.com (J.S.)

**Keywords:** rhinitis, eczema, indoor environmental factors, dampness, renovation, parental smoking

## Abstract

Asthma, rhinitis, and eczema are becoming increasingly prevalent among children in China. Studies have shown that the perinatal period is critical and impacts children’s health. However, research on the associations between perinatal factors and childhood allergic diseases in China are few. We investigated 7366 children of ages 0–8 years old. The childhood asthma and allergies were surveyed by distributing questionnaires, modelled after Dampness in Buildings and Health (DBH) study in Sweden and had been validated. To determine the prevalence of the allergic conditions, explore for potential confounders, and analyze the associations between the allergies and the home environmental factors, chi-square test and binary logistic regression models (enter method) were employed. The prevalence of children’s doctor-diagnosed asthma, rhinitis, and eczema were 4.4%, 9.5%, and 39.1%, respectively. After adjusting for sociodemographic factors, the negative effect of dampness/humidity on children’s health became more obvious, with odd ratios (aORs) of up to 1.70 (95% confidence interval (CI): 1.12–2.57) for doctor-diagnosed asthma (DDA), 2.12 (95% CI: 1.38–3.25) for doctor-diagnosed rhinitis (DDR) and 1.79 (95% CI: 1.46–2.21) for doctor-diagnosed eczema (DDE). With parental smoking, aORs of up to 4.66 (95% CI: 1.99–10.92) for DDA and 1.74 (95%: 1.00–3.02) for DDE. Renovation exhibited aORs of up to 1.67 (95% CI: 1.13–2.47) for DDR. Although they showed no significant associations with some of the health outcomes, contact with animals, in general, were risk factors for the allergic conditions. Generally, the indoor environmental factors around the perinatal period were significant risk factors for the doctor-diagnosed allergic conditions discussed in this study.

## 1. Introduction

The prevalence of childhood asthma and other allergies and their associated cost of management is rising globally [1]. This worldwide increased frequency of these allergic diseases, especially asthma, is also observed in China [2] and is a significant concern. Even though there are highly observable socioeconomic and health burdens associated with asthmas and other allergies on the population of Asia, very few investigations have concentrated on these conditions. Studies conducted in China mainly concerned the association between children’s allergic diseases and the home environmental and lifestyle-related factors [3]; however, few focused on the perinatal period, defined as the stage of child development beginning from the start of pregnancy to the end of the child’s first year of life.

Eczema, the most common childhood inflammatory skin disease, is commonly associated with other atopic manifestations [4], most of which have been linked with many perinatal factors [5]. Epidemiologic studies conducted globally have claimed associations between subsequent childhood allergies and perinatal encounter with home environmental agents. However, results from these studies appear to be inconsistent. More recently, studies [6,7] dealing with this critical period of childhood development in China have also showed positive associations between allergies and indoor environmental factors. It was suggested a link exists between the prevalence level of children’s allergic diseases in Chinese cities and the climatic regions, with a lower prevalence in cold sub-humid or dry climate [8]. Because of these aforementioned reasons, it is necessary to gain a better understanding of the associations between allergic conditions and the factors influencing them. Finding the associations between perinatal indoor environmental factors and children’s allergies will reference the prevention of childhood allergic diseases. Investigations on the association between these factors and children’s allergies in China will give data support for Chinese women’s pregnancy guidance and enrich the database of perinatal home factors and children’s allergies. This information will help health care administrators understand the magnitude of these health issues and appropriately support the planning and allocation of resources for preventive measures.

Therefore, a systematic investigation entirely devoted to indoor factors and children’s allergies was conducted with the primary aims of (1) determining the prevalence of childhood allergic conditions in Tianjin and (2) evaluating the associations between exposure to the home environmental factors (dampness/humidity, parental smoking, contact with animals, and renovation) around the perinatal period and future childhood asthma and allergic diseases in Tianjin. We hypothesize that the doctor diagnosed allergies are associated with perinatal exposure to indoor environmental factors.

## 2. Methods

### 2.1. Study Population

This work is part of the China Child Health and Home (CCHH) study at Tianjin, which was performed from 2013 to 2016 [9]. We surveyed children’s health and home environment through day care centres and elementary schools. Randomly selected kindergarten and elementary school were invited. The childhood asthma and allergies were surveyed by distributing questionnaires with the help of teachers to parents who completed and returned them. The questionnaire was formulated from the Dampness in Buildings and Health (DBH) study in Sweden [10], which was applied in several countries and cities [11]. The questions were modified to fit the Chinese context. The questions on children’s asthma and allergy were identical to those in the ISAAC (International Study on Asthma and Allergy of Child) study [12]. The questionnaire has been tested with optimum validity and reliability in previous Chinese studies [13]. The English and Chinese versions of the entire questionnaire are shown in Appendix A. This paper focuses on perinatal home environmental factors and children’s health outcomes. The Research Office at Tianjin University, Tianjin, China approved the study and consent procedure.

### 2.2. Explanatory Variables

The main perinatal home environmental factors discussed in this work are dampness/humidity, parental smoking, contact with animals, and indoor renovation. Dampness/humidity was assessed by questions such as “In the child’s residence of birth, did you notice visible mold or damp stains on the floor, walls or ceilings; detached or discolored/blackened floor coverings, condensation of moisture?” “Did anyone of the odors (stuffy, unpleasant, pungent, moldy, and others) occur in the child’s birth residence?” Questions such as “Did mother or father smoke during pregnancy or child’s first year of life?” were used to evaluate parental smoking. Similar questions were asked for contact with animals and renovation relative to the mother’s pregnancy or child’s first year of life. Contact with animals (cats and dogs) was defined as a child encountering animals during the mother’s pregnancy or child’s first year of life.

### 2.3. Outcomes

In this study, self-reported doctor-diagnosed asthma (DDA), doctor-diagnosed rhinitis (DDR), doctor-diagnosed eczema (DDE), were used to obtain information about children’s allergic diseases. To compare the results of doctor-diagnosed allergic conditions with the current allergic status of the child, results of current wheezing, current dry cough, current rhinitis, and current eczema were added. This is very critical, especially when considering asthma and the young age of the current population. Current symptoms and allergic conditions are defined as those experienced in the past 12 months.

### 2.4. Statistical Analysis

In the statistical analysis, the sociodemographic factors, including child’s gender, age, house type, home location, family income, mother’s education level and family allergy history, were potential confounders. The crude associations between these sociodemographic factors and children’s allergic diseases were obtained from a chi-square test. The associations between the allergies and the perinatal home environmental factors were analyzed using binary logistic regression models (enter method), and the odd ratios (aOR) were calculated with adjustment for confounders. Statistical analyzes were performed with the Statistical Package for the Social Sciences (SPSS), version 26.0. Armonk, NY, IBM Corp for all analyzes, a *p*-value of <0.05 was significant.

## 3. Results

### 3.1. Participant

Seven thousand eight hundred and sixty-five (7865) families (from 10,000 who received questionnaires) participated in this investigation, returning answered questionnaires. This represented a 78% response rate. Finally, 7366 children were involved in the analysis, among whom 4247children lived in the city centre, 822 in the suburbs, and 2297 in the village or countryside. The demographic information on the responding families was analysed. Table 1 summarizes the sociodemographic information, and health outcomes. More of the children were diagnosed with eczema 39.1% than were with both asthma 4.4% and rhinitis 9.5%. Current symptoms were more prevalent than doctor-diagnosed allergic diseases. About 97% of the respondents were 3–8 years old, with over half of them aging between 6–8 years old, while a little more than half of them were males. The results also indicate that the urban population (5.6% DDA, 12.2% DDR, and 46.0% DDE) reported more allergic conditions than rural children (2.6% DDA, 4.5% DDR, and 28.0% DDE).

### 3.2. Perinatal Home Environmental Factors

Table 2 demonstrates the distribution of perinatal home environmental factors. The results show that, for the indexes under dampness/humidity at the child’s residence of birth, more respondents 25.4% experienced condensation on windows and perceived dry air 26.7% than any other. Paternal smoking and change of paints (wall coverings) were higher than maternal smoking and renovation of floor coverings, respectively. Contact with dog (10.0%) were registered more often than with cat (1.9%).

### 3.3. Sociodemographic Factors and Allergic Conditions

The chi-square test results for the association between sociodemographic factors, and health outcomes are provided in Table 3. It shows significant links between these factors and the health outcomes; therefore, the sociodemographic factors could be adjusted in the following analysis. To analyse the relationship between the perinatal home environmental factors and the health outcomes, a binary logistic regression model was used without adjustment (shown inAppendix A) and later with adjustment for the sociodemographic factors (Table 4), respectively. The binary logistic regression analyses indicate an association between all children’s allergies and the indexes of dampness/humidity. Among the indexes of parental smoking after adjustment, maternal smoking shows the strongest associations with allergic diseases in terms of aORs, having a statistically significant association with DDA and current wheeze. There were also statistically significant associations between parental smoking and DDE. After adjustments, contact with cats shows significant association with current wheeze and current dry cough in terms of aOR. Again, the data exhibit positive associations between the health outcomes and the indexes of renovation, which were strengthened after adjustments in most cases.

## 4. Discussion

This study analyzed the relationships between perinatal indoor environmental factors and children’s asthma, rhinitis, and eczema based on a questionnaire survey. The prevalence of doctor-diagnosed allergic conditions reported in this study confirms the increase in incidences of these childhood conditions that was highlighted in some studies. In 2003, a large and robust study [14] covering almost the entire China, reported an average asthma prevalence of 0.99% in North China which is lower than the average 4.4% (5.6% for urban Tianjin) reported in the current study.

The prevalence of DDA in rural, suburban, and urban Tianjin are 2.6%, 2.7%, and 5.6%, respectively. Similar trends were observed for all the other health outcomes including current symptoms, with the prevalence rates following this pattern: rural < suburban < urban. These results corroborate findings reported by Ege and associates [15], indicating an increased prevalence of asthma and allergies in industrialized cities and some protection from these allergic conditions in rural communities. As with similar observations for other cities, the development of Urban Tianjin into a manufacturing megacity with a dense population is accompanied by novel ways of life and increased intensity of environmental factors, including exposure to indoor air pollutants and other allergens. However, these rapidly changing lifestyle and environmental factors could promote the rise of the incidence and prevalence of asthma and allergies in urban areas than in rural settlements.

In the present work, the findings indicate that prevalence of diagnosed asthma in males (5.3%) was a little higher than females (3.5%). While sex hormones are important in regulating the inflammatory response in asthma, other factors are also important, particularly in children prior to puberty when sex hormone levels are minimal. Although the reason for boys being more susceptible to asthma and allergies than girls is not well established, a comprehensive review [16] conducted towards this end suggested the following possible reasons among others: (1) boys have narrower airways than girls (2) boys’ peripheral blood mononuclear cells (PBMCs) generated more IFN- in response to phytohaemagglutin stimulation than girls’ PMBCs, and (3) boy could be higher genetic predisposition to develop allergic disease than girls as evidenced by specific IgE and skin prick testing to common allergens. Our results show positive associations between dampness/humidity in early childhood home environments and all the allergic diseases. The findings corroborate a previous report [17], signifying an association between dampness and doctor-diagnosed allergic disorders. Fungi such as molds and mildews thrive in humid indoor conditions. People’s allergic sensitivity to fungi was reported, with a substantial percentage of the asthmatic and general populations characterized as having specific immunoglobulin E (IgE) for fungal allergens [18]. In addition, several epidemiological investigations on the effect of dampness/fungi in indoor environments, causing odor and perception of air dryness, have suggested a link between these factors and irritant upper respiratory symptoms, such as irritation and congestion of the throat and nose and rhinorrhea. Besides, dampness could enhance the generation of endotoxins, not traditionally associated with moisture, and whose asthmatic symptoms are found to be heightened by known agents related to humidity [19]. Fungi produce a host of irritant microbial volatile organic compounds (MVOCs) [20]. The presence of moisture or high relative humidity in indoor environments amplifies the emission of non-biologic irritants such as formaldehyde [21]. There are several sources substantiating the exacerbation of asthma and other allergies because of dampness/fungi [22].

Our results suggest strong and statistically significant associations between maternal smoking and DDA and current wheeze, with aORs of 4.66 at a CI of 1.99–10.92 and 4.16 at a CI of 1.88–9.20, respectively. In fact, the associations with maternal smoking were the strongest. Some studies have indicated that smoking, in households with a relatively high prevalence of wheezing, was non-allergic, having a relatively non-threatening diagnosis, and that parental smoking was only a trigger rather than being the origin of the disease. Recent studies have documented stronger associations between maternal smoking during pregnancy (MSP) and asthma development. In this current study, the association of asthma with maternal smoking during the early stage of a child’s life was robust and strengthened after adjustment. However, the association between maternal smoking and allergies may be due to nicotine-induced negative effects caused by MSP, which include adverse effects on the child’s future immune functioning, and may also be epigenetically detrimental [23]. Hence, these allergic conditions in the later stages of a child’s life could manifest the damages caused to the fetus by maternal smoking.

Parental smoking appears to be a controversial risk factor for allergic rhinitis. Two studies [3,24] showed that maternal smoking is a risk factor for allergic rhinitis. However, the associations between the parental smoking indexes during early childhood and both DDR and current rhinitis in this study were not significant and weak in most cases in terms of aORs even after adjustment. This agrees with another study [25] which reported no significant relationship between parental smoking and allergic rhinitis. Although the size of the sample used in this study is fairly large, larger studies with more definite criteria for diagnosis are needed for reaching unambiguous conclusions. On the other hand, our findings show parental smoking to be significantly associated with DDE. Similar results were reported by Singh et al. [24]. Overall, the mother smoking indexes exhibited more profound associations than the father smoking indexes. This is probably due to the longer time children spent with their mothers, reflecting a general pattern from studies conducted globally [26].

Contact with animals did not show statistically significant associations with most of the doctor-diagnosed allergic diseases and current symptoms except for current dry cough and current wheeze (cat). Nevertheless, they generally show positive association with the health outcomes. This is in agreement with another study [27], which reported that contacts with animals were risk factor for asthma and allergy. However, controversies persist regarding the association of animals with allergic diseases. The disparities in results for individual animals with different allergic diseases, and in different locations, are well-documented. Therefore, although our findings indicated that contact with animals had positive associations with all the health outcomes, a further study on the effect of perinatal contact with animals especially in the context of biological specific mechanisms is warranted. Fortunately, efforts are already underway to isolate cowshed bacterial flora for examination [28].

Binary logistic regression tests (after adjustments) show positive statistically significant associations between the renovation of floor and wall materials (in child’s and parents’ rooms) and the allergic diseases, especially with rhinitis. The link between renovations and the allergies remained in most cases in terms of aORs, which were up to 2.19 at a 95% CI of 1.34–3.59. These results are similar to those found by one investigation [29] which indicated that renovation (redecoration) was a risk factor for respiratory symptoms and asthma in children. It is well-established that renovation materials produce substances suggested to be the culprits for exacerbating or triggering allergies.

### 4.1. Allergy Prevention and Control

Several studies have focused on the prevention and control of asthma and allergies thereby increasing our understanding of these predicaments. These studies mostly concentrate on childhood asthma and allergies and despite the wealth of knowledge gained, obstacles still exist in the development of safe and effective techniques against asthma and allergies. Since there is no proof that avoidance of allergens is beneficial, home dwellers must focus on the reduction of exposure to allergens by limiting contact with their sources or factors that promote their production. For example, the presence of dampness and, in effect, molds in residential building should be limited. This can be done by lowering indoor humidity, and preventing mold and mildew build up inside the home. These measures are achievable via several schemes including improvement of air flow through rooms, using exhaust fans, opening windows, fixing any leaks, and ensuring rainwater drains away from your house.

Tobacco smoke contains over 7000 toxic agents, including nicotine and more than 70 cancer-causing chemicals, making it a major asthma trigger. Smoking parents are advised to refrain from smoking in their homes, whether or not their children are around, due to the dangers of first- and second-hand smokes. In fact, pregnant women should abstain from smoking completely. In addition, the connection between asthma and allergies and animal contact is a contentious topic that has prompted multiple debates. Growing up in the rural regions, on the other hand, tends to be one of the most significant factors in protecting against asthma and allergies. As a result, a thorough understanding of rural environmental factors and how they affect early immune development could aid in the development of effective primary prevention strategies. Additionally, since building materials (floor and wall coverings) are suspected allergens and irritants, it is better to avoid home repairs during the perinatal period, since this is a risk factor for asthma and allergies.

### 4.2. Limitation and Strength

The method of data collection in this study, which was questionnaire-based and relied on the parents’ responses, may generate recall bias. Further, the use of self-reported doctor-diagnosed conditions may not be independent of recall bias and individual differences in symptom perception, although they may be routinely medically diagnosed cases. In addition, the independence and accuracy as regards questionnaire-generated responses are usually influenced by the background of the respondents [30].

Although cross sectional studies are generally preferred because of its relatively low cost, simplicity of design, and rapid conduct, it may be prone to bias as stated above and may not be representative of the population. Moreover, because data on each participant are recorded only once it would be difficult to infer the temporal association between the home perinatal factors and the doctor-diagnosed diseases. Therefore, a longitudinal study is required to strengthen this work. Without follow-up data gathering, it is difficult (if not impossible) to infer that the allergies were caused by the indoor perinatal factors discussed in this work. Furthermore, subsequent data collection will ensure description of trends in the associations between the allergic conditions and the home factors overtime. Further, prenatal exposure to outdoor pollution like for example, PM2.5 and PM10, need to be considered in future studies.

However, the associations between the different factors and the allergic diseases cannot be accounted for by recall or selection bias. The assessment and validation of the questions in this survey have been conducted and validated in many studies as stated above. Further, there were many participant families with a high response rate of 78%, and no previous knowledge on the information of the participant families. For these reasons, recall bias was significantly minimized.

## 5. Conclusions

This work systematically investigated the association between perinatal exposure to several indoor environmental factors with asthma and allergy in China’s Tianjin area. The findings indicated an increased prevalence in allergic conditions, but the prevalence was higher in the urban areas than was in rural communities. In general, the perinatal indoor factors showed a positive association with asthma and allergies, especially dampness and maternal smoking, which were statistically significant and strengthened after adjustment. Our research reinforces the hypothesis that perinatal exposures to indoor factors are associated with the subsequent development of childhood asthma and allergies. These results accentuate the need to formulate public education guidelines and policies geared towards reducing exposure to indoor environmental factors for Chinese pregnant women and infants. These health instruments will significantly and effectively decrease the risk of childhood allergic diseases in China.

As China experiences rapid economic development growth, its urban inhabitants will face considerable health challenges due to environmental factors such as indoor air pollution. Therefore, lessening exposure to indoor environmental factors helps abate the prevalence of childhood allergic diseases. Effective measures to curtail exposure to these factors include increasing home ventilation, reducing tobacco smoke, and avoiding dampness and mold, especially around the perinatal period. However, further investigations are needed to examine evidence of a direct biological background of the role played by home environmental factors in the pathogenesis of these allergic conditions to confirm the results.

## Figures and Tables

**Table 1 ijerph-18-04131-t001:** Demographic information, and health outcomes of investigated children.

Demographic Factors	Total (%)
**Age(years)**	
0–2	225 (3.0)
3–5	3238 (44.0)
6–8	3903 (53.0)
**Gender**	
Male	3780 (52.0)
Female	3499 (48.0)
**House Types**	
Bungalow	2306 (33.0)
Apartment	4709 (67.0)
**Family Total Income (Thousand RMB/year)**	
<50	2639 (41.0)
50–100	1863 (29.0)
100–200	1381 (21.0)
>200	572 (9.0)
**Mother Education Level**	
≤Senior high school	4115 (57.9)
Undergraduate	2248 (31.6)
Graduate	746 (10.5)
**Home Location**	
Rural	2297 (31.2)
Suburban	822 (11.2)
Urban	4247 (57.7)
**Family Allergy History**	
Yes	948 (14.1)
No	5790 (85.9)
**Health Outcomes**	
DDA	302 (4.4)
Current wheeze	333 (4.9)
Current dry cough	937 (13.6)
DDR	636 (9.5)
Current Rhinitis	2002 (29.8)
DDE	2624 (39.1)
Current eczema	998 (14.9)

DDA = doctor-diagnosed asthma, DDR = doctor-diagnosed rhinitis, DDE = doctor-diagnosed eczema.

**Table 2 ijerph-18-04131-t002:** Distribution of perinatal environmental factors.

Factor	n (%)
**Dampness/mold**	
Visible mold or damp stain	541 (14.1)
Carpet peeling up or discolored	401 (10.4)
Flooding damage	225 (5.8)
Condensation on windows	985 (25.4)
Suspected moisture problem	626 (16.1)
**Humidity/odor**	
Stuffy smell	919 (23.2)
Unpleasant smell	498 (12.7)
Pungent smell	201 (5.2)
Mildew smell	278 (7.1)
Perceived dry air	1049 (26.7)
Perceived humid air	619 (15.8)
**Parents smoking**	
FSP	2807 (41.7)
FSF	3108 (46.0)
MSP	82 (1.7)
MSF	88 (1.8)
**Contact with animals**	
Cat	137 (1.9)
Dog	727 (10.0)
**Renovation**	
FCR	279 (4.2)
FPR	268 (4.0)
PCR PPR	481 (7.3) 488 (7.2)

FSF = father smoking during the child’s first year of life, FSP = father smoking during pregnancy, MSP = mother smoking during pregnancy, MSF = mother smoking during the child’s first year of life, FCR = floor change in child’s room, FPR = floor change in parents’ room, PCR = paint/wall in child’s room, PPR = paint/wall in parents’ room.

**Table 3 ijerph-18-04131-t003:** Associations between sociodemographic factors, and children’s allergic diseases using the Chi-square test.

Factor	DDA n (%)	Current Dry Cough n (%)	Current Wheeze n (%)	DDR n (%)	Current Rhinitis n (%)	DDE N (%)	Current Eczema n (%)
**Age (years)**							
0–2	7 (3.3)	33 (15.4)	18 (8.5)	8 (4.0)	63 (30.9)	105 (50.7)	63 (30.3)
3–5	131 (4.3)	488 (16.0)	172 (5.8)	258 (8.7)	960 (32.3)	1288 (43.3)	494 (16.7)
6–8	164 (4.5)	416 (11.4)	143 (4.0)	370 (10.4)	979 (27.6)	1231 (34.9)	441 (12.5)
*p*	0.65	**0.000**	**0.000**	**0.002**	**0.000**	**0.000**	**0.000**
**Gender**							
Male	186 (5.3)	501 (14.1)	206 (5.9)	378 (11.0)	1083 (31.4)	1403 (41.1)	516 (15.0)
Female	112 (3.5)	428 (13.0)	125 (3.9)	251 (7.8)	903 (28.3)	1191 (37.1)	472 (14.8)
*p*	**0.000**	0.19	**0.000**	**0.000**	**0.005**	**0.001**	0.81
**House types**							
Bungalow	59 (2.7)	205 (9.4)	65 (3.1)	104 (4.9)	408 (19.6)	603 (28.9)	261 (12.4)
Apartment	228 (5.1)	705 (15.5)	250 (5.6)	516 (11.6)	1550 (34.5)	1976 (44.1)	713 (16.0)
*p*	**0.000**	**0.000**	**0.000**	**0.000**	**0.000**	**0.000**	**0.000**
**Family Total Income (Thousand RMB/year)**							
<50	83 (3.4)	271 (11.0)	80 (3.3)	172 (7.2)	555 (23.3)	744 (31.3)	276 (11.6)
50–100	82 (4.6)	286 (15.7)	105 (5.9)	188 (10.6)	581 (32.7)	767 (43.1)	305 (17.2)
100–200	74 (5.6)	223 (16.6)	88 (6.6)	155 (11.6)	503 (37.9)	630 (47.5)	229 (17.2)
>200	36 (6.5)	78 (13.9)	31 (5.6)	65 (11.7)	202 (36.3)	260 (46.8)	101 (18.1)
*p*	**0.001**	**0.000**	**0.000**	**0.000**	**0.000**	**0.000**	**0.000**
**Mother Education Level**							
≤Senior high school	112 (2.9)	402 (10.4)	131 (3.5)	260 (7.0)	826 (22.3)	1135 (30.7)	433 (11.8)
Undergraduate	144 (6.7)	388 (17.8)	144 (6.6)	275 (12.8)	823 (38.1)	1050 (48.6)	411 (18.9)
Graduate	39 (5.4)	131 (17.9)	53 (7.2)	90 (12.5)	316 (43.4)	388 (53.7)	136 (18.7)
*p*	**0.000**	**0.000**	**0.000**	**0.000**	**0.000**	**0.000**	**0.000**
**Home Location**							
Rural	56 (2.6)	180 (8.7)	68 (3.4)	91 (4.5)	398 (20.3)	559 (28.0)	251 (12.9)
Suburban	19 (2.7)	87 (11.6)	19 (2.6)	58 (8.1)	158 (21.7)	214 (29.6)	67 (9.2)
Urban	227 (5.6)	670 (16.2)	246 (6.1)	487 (12.2)	1446 (35.9)	1851 (46.0)	680 (16.9)
*p*	**0.000**	**0.000**	**0.000**	**0.000**	**0.000**	**0.000**	**0.000**
**Family Allergy History**							
Yes	116 (12.6)	191 (20.5)	102 (11.2)	225 (24.9)	458 (50.3)	540 (58.4)	260 (28.5)
No	166 (3.0)	695 (12.4)	207 (3.7)	371 (6.8)	1427 (25.8)	1962 (35.9)	680 (12.4)
*p*	**0.000**	**0.000**	**0.000**	**0.000**	**0.000**	**0.000**	**0.000**

DDA-doctor-diagnosed asthma, DDR-doctor-diagnosed rhinitis, and DDR-doctor-diagnosed eczema. Bold represents *p* value < 0.05.

**Table 4 ijerph-18-04131-t004:** Associations between perinatal factors and children’s allergic diseases using binary logistic regression with adjustments.

Factor	DDA	Current Dry Cough	Current Wheeze	DDR	Current Rhinitis	DDE	Current Eczema
	aOR (95%CI)	aOR (95%CI)	aOR (95%CI)	aOR (95%CI)	aOR (95%CI)	aOR (95%CI)	aOR (95%CI)
	No Ref 1.00	No Ref 1.00	No Ref 1.00	No Ref 1.00	No Ref 1.00	No Ref 1.00	No Ref 1.00
**Dampness (mold)**							
Visible mold or damp stain	**1.70 (1.12–2.57)**	**1.77 (1.36–2.31)**	**2.26 (1.55–3.30)**	**1.55 (1.13–2.14)**	**1.79 (1.44–2.24)**	**1.33 (1.07–1.66)**	**1.62 (1.24–2.11)**
Carpet peeling up or discolored	1.51 (0.93–2.45)	**1.70 (1.26–2.30)**	**1.80 (1.15–2.81)**	**1.55 (1.08–2.23)**	**1.64 (1.27–2.15)**	1.18 (0.92–1.52)	**1.40 (1.03–1.92)**
Flooding damage	1.18 (0.82–1.70)	**1.65 (1.12–2.42)**	**2.33 (1.40–3.89)**	**2.12 (1.38–3.25)**	**1.59 (1.14–2.22)**	1.10 (0.79–1.53)	1.11 (0.73–1.69)
Condensation on windows	1.67 (1.12–2.48)	1.22 (0.97–1.54)	**1.63 (1.16–2.29)**	**1.41 (1.08–1.84)**	**1.33 (1.12–1.60)**	**1.47 (1.24–1.75)**	**1.50 (1.20–1.88)**
Suspected moisture problem	1.09 (0.75–1.58)	1.63 (1.26–2.11)	**1.93 (1.32–2.83)**	**1.49 (1.10–2.03)**	**1.46 (1.18–1.80)**	**1.51 (1.23–1.85)**	**1.70 (1.32–2.19)**
**Humidity and odor**							
Stuffy smell	1.31 (0.86–1.99)	**1.58 (1.26–1.98)**	1.35 (0.95–1.92)	**1.51 (1.15–1.98)**	**1.64 (1.37–1.96)**	**1.67 (1.40–1.99)**	**1.91 (1.53–2.38)**
Unpleasant smell	1.27 (0.62–2.59)	**1.79 (1.36–2.35)**	**2.18 (1.47–3.21)**	**1.51 (1.09–2.10)**	**1.72 (1.37–2.16)**	**1.46 (1.18–1.82)**	1.25 (0.95–1.65)
Pungent smell		**1.74 (1.15–2.64)**	**2.78 (1.63–4.76)**	**1.87 (1.16–3.01)**	**1.89 (1.34–2.68)**	**1.51 (1.08–2.12)**	**2.08 (1.40–3.10)**
Mildew smell	1.58 (0.92–2.70)	**2.04 (1.45–2.86)**	**1.81 (1.08–3.04)**	**1.64 (1.09–2.48)**	**2.33 (1.73–3.13)**	**1.69 (1.27–2.26)**	**2.27 (1.62–3.16)**
Perceived dry air	1.24 (0.87–1.76)	**1.55 (1.25–1.93)**	1.39 (0.99–1.96)	**1.32 (1.01–1.72)**	**1.56 (1.31)**	**1.45 (1.23–1.72)**	**1.45 (1.16–1.80)**
Perceived Humid air	1.41 (0.93–2.14)	**1.57 (1.21–2.04)**	1.35 (0.89–2.05)	**1.45 (1.05–2.00)**	**1.70 (1.37–2.10)**	**1.79 (1.46–2.21)**	**1.94 (1.50–2.51)**
**Parental smoking**							
FSP	1.16 (0.88–1.53)	1.07 (0.91–1.26)	**1.45 (1.12–1.89)**	1.02 (0.84–1.25)	1.06 (0.93–1.20)	**1.20 (1.07–1.35)**	1.06 (0.91–1.25)
FSF	1.15 (0.87–1.51)	1.03 (0.88–1.21)	**1.32 (1.01–1.71)**	1.05 (0.87–1.29)	1.06 (0.93–1.20)	**1.16 (1.03–1.30)**	0.99 (0.85–1.16)
MSP	2.44 (0.92–6.46)	1.28 (0.62–2.65)	**4.66 (2.17–10.00)**	1.26 (0.52–3.08)	0.88 (0.48–1.63)	**1.74 (1.00–3.02)**	1.22 (0.61–2.43)
MSF	**4.66 (1.99–10.92)**	1.07 (0.50–2.28)	**4.16 (1.88–9.20)**	2.04 (0.92–4.52)	0.90 (0.49–1.64)	1.38 (0.81–2.35)	1.11 (0.55–2.25)
**Contact with animals**							
Cat	2.18 (0.98–4.87)	**1.74 (1.03–2.94)**	**2.49 (1.17–5.30)**	1.65 (0.85–3.19)	1.24 (0.79–1.96)	1.14 (0.75–1.73)	1.44 (0.85–2.43)
Dog	0.66 (0.37–1.16)	1.18 (0.90–1.54)	1.05 (0.66–1.65)	1.07 (0.76–1.50)	1.08 (0.88–1.34)	1.14 (0.94–1.38)	1.22 (0.95–1.57)
**Renovation**							
FCR	1.05 (0.54–2.05)	**1.52 (1.07–2.17)**	**2.19 (1.34–3.59)**	**1.53 (1.01–2.33)**	**1.35 (1.00–1.83)**	1.08 (0.81–1.45)	**1.78 (1.26–2.52)**
FPR	1.27 (0.71–2.26)	**1.54 (1.11–2.15)**	**1.85 (1.12–3.08)**	**1.67 (1.13–2.47)**	0.94 (0.81–1.10)	1.27 (0.98–1.63)	**1.71 (1.23–2.38)**
PCR	1.34 (0.84–2.15)	**1.41 (1.07–1.87)**	**1.70 (1.12–3.08)**	**1.51 (1.08–2.09)**	**1.35 (1.08–1.70)**	0.94 (0.75–1.18)	1.30 (0.97–1.71)
PPR	1.06 (0.71–1.58)	**1.50 (1.16–1.95)**	1.39 (0.91–2.12)	**1.41 (1.03–1.94)**	0.94 (0.84–1.05)	0.97 (0.86–1.09)	1.29 (0.99–1.68)

DDA = doctor-diagnosed asthma, DDR = doctor-diagnosed rhinitis, DDE = doctor-diagnosed eczema, FSF = father smoking during the child’s first year of life, FSP = father smoking during pregnancy, MSP = mother smoking during pregnancy, MSF = mother smoking during the child’s first year of life, FCR = floor in child’s room, FPR = floor in parents’ room, PCR = paint/wall in child’s room, PPR = paint/wall in parents’ room. Odd ratios of the variables were calculated in binary logistic regression. Bold represents *p*-value < 0.05. Adjusted for age, gender, house types, family total income, family allergy history, home location, mother’s education level.

## Data Availability

The data presented in this study are available in the Appendix A.

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
