# Peer review of "Prevalence of Childhood Asthma and Allergies and Their Associations with Perinatal Exposure to Home Environmental Factors: A Cross-Sectional Study in Tianjin, China"

_ijerph, 2021, doi:10.3390/ijerph18084131_

Round 1

Reviewer 1 Report

Current manuscript by Dr. Yuexia Sun reports a cross-sectional study conducted in Tianjin, China involving 7366 children focusing on the relation between perinatal factors and the occurrence of allergic diseases: asthma, rhinitis and eczema.

Authors compared various aspects ranging from socioeconomic to environmental factors with the incidences of “doctor-diagnosed asthma (DDA), doctor-diagnosed rhinitis (DDR), doctor-diagnosed eczema (DDE)” along with “current wheezing, current dry cough, current rhinitis, and current eczema”. The questionnaire used was identical to the ISAAC (International Study on Asthma and Allergy of Child) study.

The manuscript is well written with methods, results and discussion being moderately comprehensive. The study is a good start, and more analyses should be employed on the dataset to identify other critical trends. Authors should also plan on longitudinal study, following on some of the factors.

Authors are requested address the following suggestions:

  1. Authors are requested to comment on the possible outcome from a longitudinal approach. Obviously several factors cannot be accounted for, but more convincing inferences can be observed. Please consider including a statement on this in the “4.1 Limitation and strength” section of Results.
  2. Table 1, for simplicity, authors may consider changing the percentage for 0-2 years to 3.0% instead of 3.1. Similar editing is suggested for Family Total Income (Thousand RMB/year) too.
  3. Please elaborately discuss the association between smoking and the allergies. Particularly, include the product use pattern if available and whether any new electronic nicotine delivery product use was reported.
  4. Male gender seems to be more associated with allergic diseases which should be expanded in the discussion by the authors.
  5. An important factor excluded from study was the use of illicit drugs and the allergic diseases. Please comment if any such information is available.
  6. Authors may also consider any published information on the environmental pollution levels, and may attempt to correlate the same with allergies.
  7. When considering the environmental factors, did authors enquire about the educational institution ambiance and associated factors? Children spend substantial amount of time in these establishments, and this may influence the results.
  8. Table 3, it is interesting to see that almost all of the factors are significantly associated. Authors may alter the data analysis approach to identify more convincing associations. The current simple chi-square may not be the most appropriate approach and a different statistical analysis may be useful. Please comment.
  9. Authors may consider including a section on potential suggestions to control the allergies.
  10. An important aspect excluded from the study is the height of the children that greatly determines exposure to some environmental factors. Authors may consider some discussions on this.

Reviewer 2 Report

This is a well researched and written study which adds to the literature.

In regards to questionnaires, information is more likely to be influenced by respondents background factors and should be addressed as a limitation (Miao Q, et al. BMC Pediatr 2020; 20: 259).

The authors grouped dog and cat exposure together, Some studies have shown different effects of individual animal exposure. For instance, Li et al. (Front Pediatr 2020; 8: 192) showed that early life exposure to dogs significantly decreased the risk of atopic dermatitis at age 5 in a prospective birth cohort. While allergic family history significantly reduced the risks to owning a cat or dog (Allergy 2008; 63: 191-8).

Commonly used questions, like "are there pets at home" can lead to erroneous interpretation of real risk of exposure in epidemiological studies (Liccardi et al. Eur Ann Allergy Clin Immunol 2016; 48: 61-4).
